# Building Consensus on the Point-of-Care Ultrasound Skills Required for Effective Healthcare Service Delivery at District Hospitals in South Africa: A Delphi Study

**DOI:** 10.3390/ijerph20237126

**Published:** 2023-11-30

**Authors:** Pierre-Andre Mans, Parimalaranie Yogeswaran, Oladele Vincent Adeniyi

**Affiliations:** 1Department of Family Medicine, Cecilia Makiwane Hospital, Mdantsane, East London 5201, South Africa; oadeniyi@wsu.ac.za; 2Department of Family Medicine and Rural Health, Faculty of Health Sciences, Walter Sisulu University, Mthatha 5117, South Africa; pyogeswaran@wsu.ac.za; 3Department of Family Medicine, Mthatha Regional Hospital, Mthatha 5100, South Africa

**Keywords:** curriculum, Delphi study, district hospitals, point-of-care ultrasound, South Africa

## Abstract

Background: Despite the widespread availability of ultrasound machines in South African district hospitals, there are no guidelines on the competency in point-of-care ultrasound (POCUS) use required by generalist doctors in this setting. This study aimed to define the required POCUS competencies by means of consensus via the Delphi method. Methods: An online Delphi process was initiated in June 2022, using the existing American Academy of Family Physicians’ ultrasound curriculum (84 skillsets) as the starting questionnaire. Panelists were selected across the country, including two from district hospitals in each province and two from each academic family medicine department in South Africa (*N* = 36). In each iterative round, the participants were asked to identify which POCUS skillsets were essential, optional (region-specific), or non-essential for South African district hospitals. This process continued until consensus (>70% agreement) was achieved on all of the skillsets. Results: Consensus was achieved on 81 of the 84 skillsets after 5 iterative rounds (96.4%), with 3 skillsets that could not achieve consensus (defined as <5% change over more than 2 consecutive rounds). The final consensus identified 38 essential, 28 optional, and 15 non-essential POCUS skillsets for the South African district hospital context. Conclusions: The list of essential POCUS skillsets provided by this study highlights the predominance of obstetric- and trauma-based skillsets required for generalist healthcare workers in South African district hospitals. The findings will require priority setting and revalidation prior to their implementation across the country.

## 1. Introduction 

There has been more than a decade of massive interest in point-of-care ultrasound (POCUS) as an extension of clinical examination [1,2]. POCUS allows the clinician to have a real time means to appreciate otherwise hidden anatomy by using sound waves—leading to the point that it is regarded as ‘the stethoscope of the future’ [3]. It has been described as a tool for improving patient outcomes [4] and has been found to improve patient satisfaction [5]. POCUS has also been identified as a potential means to close some of the gaps in global healthcare inequality, having the potential to improve patient access to diagnostic imaging and correct diagnosis in low-to-middle income countries (LMIC) [6]. This emerging evidence has led to extensive government and private investments in expanding POCUS use and availability in LMIC. One of the most recent developments was the Bill and Melissa Gates Foundation’s donation of 1000 handheld POCUS devices for use in sub-Saharan Africa [7]. 

Several medical specialties in high-income countries have identified their own POCUS curricula, comprising skillsets considered essential in their field of medicine [8,9,10]. Apart from the field of emergency medicine, there is no clear guidance on the specific POCUS skillsets required in sub-Saharan Africa [11]. This may be due, in part, to the diverse contexts that require a unique set of POCUS skills, but also because, in LMIC, most healthcare services are provided by generalists [12]. Few studies have attempted to identify the POCUS skillsets needed by generalists, and those that exist vary greatly from one context to another, with the majority conducted in developed countries [5,13,14,15]. The existing publications describing generalist POCUS skillsets in sub-Saharan Africa vary greatly in content and uptake [16,17,18,19,20].

This study aims to identify the POCUS skillsets required in South African district-level hospitals by generalist medical practitioners. District hospital services are provided by generalists, who are yet to undergo any formal specialist training at academic hospitals. Increasingly, there are also family physicians being employed at this level who also work as generalists in this context. In addition, considering the diverse geographical and socio-economic settings in South Africa, our objectives were to identify which ultrasound skillsets would be considered essential, optional (region-specific), and non-essential in South African district hospitals. The findings of this study could potentially inform the training of doctors working at this level of healthcare in the country. Also, these findings may contribute to the development of a national policy on POCUS and a teaching curriculum for the department of family medicine in South Africa.

## 2. Methods

### 2.1. Study Design

We adopted an online Delphi process as a suitable design for this study, because of the geographic and logistical challenges of assembling a group of suitable experts from across the country. The online Delphi allowed the participants to complete the questionnaire at their convenience, making it very advantageous for this study. The Delphi technique was originally developed by the Rand Corporation in the 1950s as an instrument to predict future technological advances. Subsequently, it has emerged as a useful approach for developing new concepts, revealing expert consensus on any topic. The process facilitates expert consensus through multiple rounds (iterations) of anonymous responses, providing sequential feedback of the group’s response to all items not achieving consensus [21,22]. The Delphi process is ideal, since anonymity allows all participants’ responses to be equally weighted, and the facilitated feedback allows each participant to review their response in the light of the group’s response. The addition of comments allows the participants to motivate their responses with a view to influencing other participants, or to be guided by other participants’ opinions to change their responses in successive rounds. Successive rounds will continue until a predetermined level of consensus is achieved for each item. We decided to set consensus at 70%, in keeping with the existing literature [22,23]. In addition, we decided to remove all skillsets that did not prompt a significant change in opinion (<5%) over two or more rounds, designating them as skillsets for which consensus could not be achieved. Although this last aspect is not typically part of a Delphi design, we opted to include it to streamline the process, and we included it in the published protocol [24]. The reporting for this study adheres to the CREDES (Guidance on Conducting and Reporting Delphi Studies) guidelines [25].

Although the Delphi method relies on expert opinion, the process is methodologically considered to be a mixed-method study, since the participants’ responses are quantitatively assessed and statistically reviewed by all participants after each round. To facilitate the ease of use and ensure the anonymity of responses, we used the online Welphi (Lisbon, Portugal) software to run the online Delphi process. 

### 2.2. Survey Development

Given the diverse range of conditions evident in South African district hospitals, the list of POCUS skillsets had to be extensive. Looking at the existing generalist POCUS publications from sub-Saharan Africa, the varying scope and extent were seen as major limitations. We opted to start with the POCUS skillsets identified in the American Academy of Family Physicians (AAFP) curriculum, since this is the most extensive generalist POCUS publication [24,26]. These 93 skillsets and their descriptions, along with a literature reference for each skillset, formed our starting questionnaire. The skillsets were subdivided into ten clinical domains (as shown in Table 1) that would each become a separate page in the online Delphi questionnaire (see Appendix A). 

In the original AAFP curriculum, four skillsets were duplicated in different domains, and the POCUS protocols included skillsets mentioned individually in their respective domains. We maintained this to assure academic integrity. These are as follows: the detection of a pericardial effusion (in both cardiac and trauma domains), the detection of a pneumothorax and hemothorax (in both the pulmonary and trauma domains), and, lastly, the identification of a foreign body (in both the musculoskeletal and procedural domains). The following five protocols are all compilations of skillsets that are individually described [27,28,29,30]: extended focused assessment with sonography in trauma (EFAST), rapid ultrasound in shock and hypotension (RUSH), bedside lung ultrasound in emergencies (BLUE), cardiopulmonary limited ultrasound exam (CLUE), and fundal/epileptical/eccentric/decidual reaction size >4 mm (FEEDS)The absolute number of POCUS skillsets is 84 when the duplications and protocols are removed.

In anticipation of the initial list of POCUS skillsets not being exhaustive, we invited the participants to recommend any additional POCUS skillsets in the first round, based on their experience of the local disease burden. These additional skillsets will be reviewed for duplication and included from the second round onward in all iterations, until consensus is achieved. The selection of POCUS skillsets in this study is limited to the existing published resources and participants’ suggestions, to reduce potential bias. The questionnaire allows the participants to select one of the following three options for each skillset: essential (skillsets considered essential for day-to-day practice), optional (skillsets that may be required in some contexts but will be region-specific), and non-essential (skillsets that should not be performed at a generalist level in South African district hospitals). 

### 2.3. Panel Recruitment

There are no existing guidelines on what ultrasound skillsets are required in South African generalist healthcare. In the absence of a clear qualification or inclusion criteria, we opted for peer nomination of participants. To ensure inclusivity, 9 of the 10 medical schools with an established postgraduate residency program in family medicine were included in the study. A total of 2 family physicians from each academic department of family medicine (*n* = 18) who work with POCUS in patient evaluation were nominated by their respective heads of department. Prior to recruitment, a written request was sent to the various heads of department (HODs) of family medicine to nominate these participants based on their POCUS proficiency in clinical practice. Given that most of the medical schools are situated in urban areas, the representatives of doctors drawn from the rural district hospitals gave diversity in the experience and exposure of the panel members. A total of 2 medical doctors were drawn from the district hospitals in each of the 9 provinces (*n* = 18) via nomination from the Rural Doctors Association of South Africa (RUDASA). The participants were nominated as POCUS experts by their peers based on perceived proficiency with the use of ultrasound in patient evaluation in their respective district hospitals, without any limitation on the level of training or years of experience. The nominations were grouped per province and the two doctors with the most years of POCUS experience in each province were selected to be participants in this study. 

Our participants were thus equally weighted between academic and geographic representation, with a total of 36 participants assembled for the study. Panel selection based on the existing structures of the various academic departments and the RUDASA provincial forums was intentionally implemented to reduce the risk of selection bias. 

### 2.4. Ethical Clearance 

Ethical approval was obtained from the Walter Sisulu University Research, Ethics, and Biosafety Committee (Reference: 065/2021). All participants completed a written informed consent form and were allowed to withdraw from the study at any point of the Delphi process. Anonymity of responses was maintained throughout the study. The entire study process followed the Helsinki Declaration and Good Clinical Practice Guidelines. 

### 2.5. Patient and Public Involvement

The respective heads of each postgraduate department of family medicine and the national and provincial RUDASA groups were all involved in the selection process of participants. The participants were allowed to suggest additional POCUS skillsets and were allowed to comment and select their respective opinion on all of the listed POCUS skillsets. 

### 2.6. The Delphi Process 

An initial introductory email explained the nature of the study, accompanied by an online consent form and the first round of the questionnaire. We sent weekly electronic reminders to all participants who had not yet completed the questionnaire. After two weeks, all participants who had not yet completed the questionnaire received twice-weekly individual reminders. These were carried out via either voice calls or WhatsApp text messages. The twice-weekly reminders continued until all of the participants had either completed the round or had indicated that they would prefer to withdraw from the study. This process of recurrent rounds (iterations) continued until all of the skillsets had either achieved consensus or had been removed as items that could not achieve consensus. 

A total of five rounds were needed to finalize the Delphi process. The durations of the individual Delphi rounds varied, as there was a need to ensure that all participants had either responded to or formally withdrawn from the study before a new round began. Each new round started within 24 h of the last response in the previous round. 

#### 2.6.1. Round 1

The Delphi process started on 5 August 2022. The participants were presented with the 93 skillsets across the 10 clinical domains, with a literature reference for each skillset. The participants were required to select which skillsets they deemed essential, optional, and non-essential, and comment anonymously on why they had made their selection. The participants were invited to recommend any other POCUS skillsets that were not included in the starting questionnaire. The results were collated and all skillsets that achieved consensus were removed. The first round closed on 28 August 2022. 

#### 2.6.2. Round 2

On 29 August 2022, the participants were presented with the second-round questionnaire showing all skillsets that had not achieved consensus in the previous round, together with the group’s statistical responses and their comments. The participants could review their response in light of the other responses and comments, and then choose to change their selection or add comments motivating their selection. The final response for this round was received on 2 October 2022, and again, all skillsets that achieved consensus were removed. 

#### 2.6.3. Round 3

On 3 October 2022, the participants were presented with the third-round questionnaire showing all skillsets that had not achieved consensus, together with the group’s statistical responses and comments. The participants were asked to categorize the remaining skillsets as essential, optional, and non-essential. This round closed on 2 November 2022 and all skillsets that achieved consensus were removed. In addition, any skillset that did not show more than 5% change from the first round was removed for not achieving consensus. 

#### 2.6.4. Round 4

The fourth round started on 3 November 2022. The participants were informed of the skillsets that had achieved consensus and those that could not achieve consensus in the third round. The remaining skillsets were presented, as in previous rounds. The final response was received on 29 November 2022. All items that achieved consensus were removed and the items that did not show more than 5% difference across two or more rounds were removed for not achieving consensus. 

#### 2.6.5. Round 5

The final round started on 30 November 2022. Again, an overview of the previous round was provided, identifying the skillsets that had been removed, together with the comments. The remining skillsets were presented, and the participants made their selection of essential, optional, and non-essential. The final round was completed on 15 January 2023. 

The iterative rounds had a mean duration of 31.8 days, with the final round being significantly longer (46 days), owing to the festive season and associated work holidays. 

### 2.7. Data Analysis

All participants’ responses were captured online using Welphi software, which produced anonymized feedback after each round. Next to each item, the group’s response was reported as a percentage. All of the data were extracted and entered into Microsoft Excel for further statistical analysis. The frequency (absolute number) and percentages were presented for the skillsets achieving consensus after each iterative round. 

### 2.8. Role of the Funding Source

The funder of the study had no role in the study design, data collection, data analysis, data interpretation, or writing of the report. Full access to the data was available to P.M., P.Y., and O.V.A. All authors accept responsibility for the decision to submit for publication.

## 3. Results

### 3.1. Participants

The study started with a group of 36 nominated participants, all of whom are medical doctors working in public sector district hospitals in South Africa and have some POCUS proficiency. The geographical breakdown favored the more developed provinces (Gauteng and Western Cape), because there are more medical schools located there. The participants had a wide range of years of experience, with ten participants reporting more than ten years of POCUS experience and eight participants having less than three years of experience. Four of the participants did not respond to this question. The majority (23 participants) had some form of formal ultrasound training, mostly in the form of short courses (Table 2). Overall, 20 of the original 36 participants completed all 5 iterative rounds (55.6%). The participants’ retention averaged at 88.9% for each round, with an average attrition of three participants per round. The total retention across the entire Delphi process was 55.6% (Figure 1). 

### 3.2. Skillsets

#### 3.2.1. Round 1

In the first round, no additional skillsets were recommended by the participants for inclusion in the Delphi. A total of 14 of the 93 skillsets achieved consensus (15.1%), with all being deemed essential by the 32 participants who completed the round. The retention rate was 88.9%.

#### 3.2.2. Round 2

In the second round, 79 skillsets remained, with 21 being able to achieve consensus (26.6%). Of these, 1 was seen as optional and 1 as non-essential, with the majority (19) being seen as essential. In total, 28 of the participants completed the second round. The retention rate was 87·5%.

#### 3.2.3. Round 3

In the third round, 58 skillsets were presented to the participants, with 21 being able to achieve consensus. A total of 3 were deemed to be essential, 11 were deemed to be optional, and 7 were deemed to be non-essential by the 26 participants who completed the round. In addition, two skillsets showed very little change (<5%) from the initial review in the first round. These two skillsets were removed due to being skillsets that could not achieve consensus. A total of 23 skillsets were removed at the end of the third round. The retention rate was 92.9% in this round. 

#### 3.2.4. Round 4

In the fourth round, 35 skillsets were presented, with 26 being able to achieve consensus. Again, only 3 skillsets were selected as being essential, 17 were selected as optional, and 7 were selected as non-essential. There was one skillset that showed a less than 5% change in the participants’ responses over the previous two rounds, which was removed due to being a skillset that could not reach consensus. Thus, 27 skillsets were removed at the end of the fourth round. In total, 22 participants completed this round, giving a retention rate of 84.6%. 

#### 3.2.5. Round 5

In the fifth round, the remaining 8 skillsets were presented to the 22 remaining participants. Six of these skillsets were selected as being essential, and two as optional. Only 20 participants fully completed the final round. The retention rate was 90.9% in this round (Figure 1). 

Thus, an average of 18 skillsets achieved consensus per round. Overall, 45 skillsets (48.3%) were identified as being essential, 30 skillsets (32%) were deemed to be optional, and 15 skillsets (16%) were deemed to be non-essential (Table 3). All of the skillsets were reviewed for duplication across the different domains. 

Consensus on the essential POCUS skillsets required for the South African district hospital context is shown in Table 4. The full list with individual skillset consensus per domain is presented in Appendix A. 

## 4. Discussion

This study had ascertained the POCUS skillsets that are required in South African district-level hospitals by generalist medical practitioners, categorizing them as essential, optional (region-specific), and non-essential based on consensus expert opinion. While emergency physicians have implemented POCUS in a few emergency units of district hospitals as adjunct to patient evaluation during resuscitation and HIV-associated tuberculosis [31,32,33,34], there has been no clarity regarding the extent of POCUS requirements as an adjunct to patient evaluation for effective healthcare service delivery by generalists at the district hospital level in the country. A recent study looked at district hospital level generalist POCUS requirements in a single Western Cape district and emphasized the need for formal POCUS training, as well as the importance of the idea that POCUS content should be locally adapted [35]. As such, this study presents a unique distillation of the skillsets obtained through several iterative rounds of consensus by panelists drawn from all of the medical schools and district hospitals across the country. The findings of this study could potentially guide the training of doctors working at this level of healthcare and inform the crafting of the curriculum for family medicine registrars in South Africa. 

Given the robust engagements of the panelists through the five iterative rounds of the Delphi, the summary document of the POCUS skillsets obtained from this study can be implemented through the South African Academy of Family Physicians and the National Department of Health. The majority (73%) of the essential skillsets were identified by the end of the first two iterative rounds. These essential skillsets were mostly in obstetrics and gynaecology, trauma, procedure- and protocol-based domains. This is not surprising, as the listed skillsets directly reflect the profile of the patients accessing the district hospitals in the country, where a large volume of obstetrics and trauma cases are seen [35]. 

It is interesting to note that a high proportion (48%) of the POCUS skillsets were categorized as being essential. This may be reflective of the increased popularity and media attention given to POCUS or the perceived utility of POCUS in managing undifferentiated patients in a generalist context. It may also be due to the health system limitations that make elective formal ultrasound diagnoses difficult to obtain. This limitation forces clinicians to conduct more POCUS for patients than may be required in other settings, where formal diagnostic ultrasound is more readily available. 

The fact that more than half of the optional and non-essential skillsets achieved consensus only in the later iterative rounds of this Delphi study reflects the diversity of the working environments of the panelists. The geographic representation of the panelists elicited diverse opinions on the various skillsets. The significant proportion (32%) of skillsets categorized as optional (region-specific) suggests variations in the use of POCUS in different regions of the country. This finding provides evidence for regional variations in the use of POCUS in South Africa, as well as the need for other countries in the sub-Saharan African region to adapt the skillsets presented in this study based on the local disease burden and resources. While access to ultrasound has grown remarkably in South Africa in recent years, it is noteworthy that there is gradual increase in access to POCUS in the rest of sub-Saharan Africa. There is some evidence that other sub-Saharan African countries follow South African guidelines and policy. For instance, the basic emergency point-of-care ultrasound (POCUS) course of the Emergency Medicine Society of South Africa (EMSSA) is offered to doctors across multiple Southern African countries [36,37].

Most of the non-essential skillsets were in the musculoskeletal and ocular clinical domains. This may reflect the specialized nature of these skillsets and the perceived need among the panelists for referral of these conditions to their respective specialists. This is in contrast with the high number of cardiac and abdominal skillsets that were seen as being essential or optional, in keeping with the global trends in many specialties to increase POCUS use in primary healthcare for these clinical domains [32,38].

The three skillsets that could not achieve consensus; (identification and grading of hydrocele, identification of urinary retention and post-void residual volume, and identification of rotator cuff injury) that were removed from the initial list were within the musculoskeletal and abdominal domains. This came as a surprise in the final consensus document, given that they are conditions that are encountered in daily clinical practice at the district hospital level in the country. These skillsets may require further investigation in future research as the use of POCUS grows in the country.

Another significant omission from the study findings was the absence of the FASH (focused assessment with sonography for HIV-associated tuberculosis) protocol [39]. This protocol was not part of the initial skillsets set out in the AAFP curriculum and was not suggested by the participants in the first iterative round, which was surprising given the endemicity of tuberculosis in the country. It is plausible that the widespread availability of other diagnostic techniques (GeneXpert MTB/Rif ultra, line probe assays, and lateral-flow lipoarabinomannan) for intensified TB identification could be the reason for the omission [40]. Similarly, the test and treat HIV policy, which was implemented in 2016, has reduced the incidence of advanced HIV disease and, consequently, the likelihood of extrapulmonary TB [41]. Nonetheless, the following essential skillsets, which all achieved consensus within two iterative rounds, will improve the diagnosis of HIV-associated TB at the district hospital level: the assessment of free fluid around the heart by sub-xiphoid view, free fluid in the abdomen, hepato-splenomegaly, pleural effusion, pneumothorax, and fine-needle aspiration/biopsy. The POCUS diagnosis of interstitial pneumonia (such as that seen with COVID-19) was also seen as non-essential. This is surprising, given the evidence of POCUS accuracy and ease of use in interstitial pneumonia [42]. This have been influenced by the lack of POCUS inclusion as part of the diagnostic algorithm set out by the South African National Institute for Communicable Disease [43].

Interestingly other much less commonly used skillsets were included as essential, such as the detection of abdominal aortic aneurysm and the detection of testicular torsion. This may be due to the perceived simplicity of application, or the significant morbidity and/or mortality if the diagnosis is missed, rather than the required incidence of use.

Overall, the consensus statement of the POCUS skillsets obtained from this Delphi process will open doors for future research at this level of healthcare in the country. The diversity of the academic and geographic background of the panelists involved in the study demonstrated inclusivity and makes the findings applicable to all South African district hospitals. This is evident in the depth and breadth of the skillsets that were deemed essential, such as the following: obstetrics (16 out of 20), trauma (4 out of 4), vascular (2 out of 3), procedural (6 out of 10), and diagnostic protocol skillsets (4 out of 5)—in keeping with the profile of the patients accessing care at the district hospitals. Within the more specialized fields, both the cardiac and abdominal domains had no skillsets deemed to be non-essential, and both had a similar percentage of skillsets seen as essential (40% and 41%, respectively). This is in stark contrast to the pulmonary, ocular, and musculoskeletal fields, which had the majority of the non-essential skillsets. Other African studies had varied areas of focus, with some having a predominance of obstetric skillsets [17] and others a predominance of abdominal skillsets, closely followed by obstetric skillsets [18,20]. There was only one African study that did not report obstetric skillsets as a major component of the generalist use of POCUS [16]. Our findings are in keeping with international trends that show an increase in POCUS use for cardiology applications [38,44,45,46]. However, our findings differ from some of the studies from high-income countries, which reveal a significant musculoskeletal focus at the generalist level [13,47]. This highlights the importance of regional variation in POCUS practice based on the local disease burden. 

With regards to the study limitations, the participants’ POCUS experience and levels of training were not standardized. The panelists comprised clinicians nominated by their peers, given that there are no existing guidelines or criteria for determining a POCUS expert in the South African generalist setting. As shown in Table 2, only half of the participants received some accredited training, such as a postgraduate diploma and Foundation for Professional Development and EMSSA POCUS courses. The remaining half received non-credentialled POCUS or informal training, which should not come as a surprise, given that skill transfer is a common practice at the district hospital level and peer-to-peer POCUS education is practiced not only in South Africa, but across the world [48,49]. There was also lack of clarity on whether the institutions where the participants were based had access to sonographers, or if the participants were active POCUS trainers. Variations in their levels of expertise and practice might have influenced their opinions on the applicability of some of the more specialized POCUS skillsets, and this could have had an impact on the final skillset selection. The nomination of participants based on perceived POCUS competence led to the inclusion of eight participants with less than four years of POCUS experience. This limited experience might have had a bearing on the opinions of these panelists, limiting their capacity to comment as an expert on the subject matter. Yet it may also reflect the actual working environment of the district hospitals. The attrition of 16 participants through the course of the Delphi is a notable limitation of this study, with a possible impact on the final selection of skillsets. The attrition rate is comparable to a similar study, yet it remains a limiting factor on the generalization of our findings [8]. Notwithstanding these limitations, this study has highlighted the skillsets required for medical practitioners working at district hospitals in the country, which serve as the first point of care for the majority of South African residents. It is, therefore, important to engage the stakeholders (South African Academy of Family Physicians, the heads of the various departments of family medicine, and the National Department of Health) on how best to implement the findings of this study. The list of 45 essential skillsets produced by our expert panel is more comprehensive than any previous publication guiding POCUS use in a generalist context. This essential list provides a combination of POCUS skillsets across multiple clinical domains. Its implementation will require a stepwise approach, with possible adaptation based on the local disease burden in each region of the country. Perhaps, an observational study on the current indications for requesting or performing ultrasound at the district hospitals in the country could guide the process of priority setting for the implementation of the essential and optional skillsets. This will also require the training of instructors at the district health and provincial level in order to scale up the POCUS competency in the country. The essential skillsets could also be weighted based on their ease of use, frequency of need, and difficulty to perform—a method applied by both van Hoving and Heller to identify the most appropriate skillsets for further training [39,50]. More studies are needed on how to scale up the POCUS skillsets at the district hospital level in South Africa. 

## 5. Conclusions

This study reports a long list of essential, optional, and non-essential POCUS skillsets for effective health care service delivery at the South African district hospital level. However, further studies are needed to revalidate and determine the priority skillsets for implementation across the country. A stepwise approach guided by the local disease burden and priority setting of each region is, therefore, recommended for the district hospital doctors in the country. The findings from this study can be adapted into the training curriculum of postgraduate students in the department of family medicine in the country. Further research is required in the form of a cost–benefit analysis on the inclusion of POCUS in training programs. While resources and disease burdens may differ in other African countries, the findings need to be suitably adapted to each country’s need for POCUS skillsets. 

## Figures and Tables

**Figure 1 ijerph-20-07126-f001:**
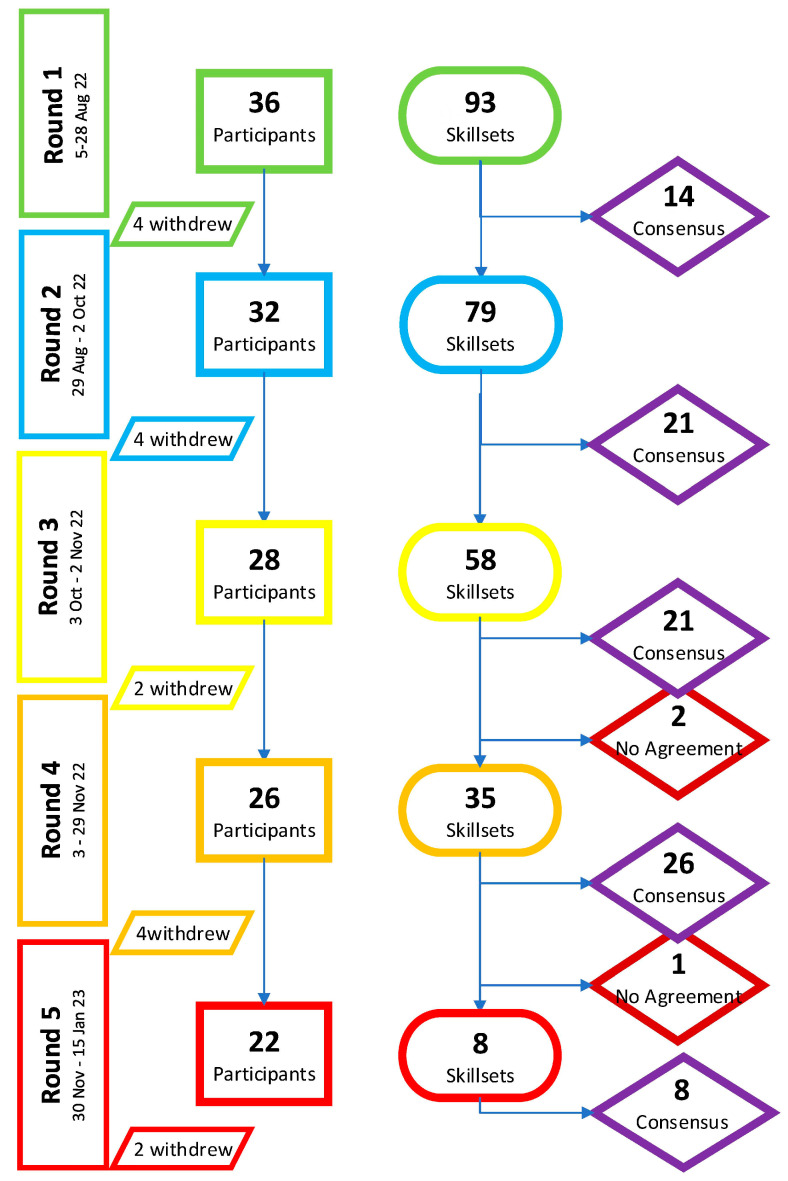
Flowchart of the Delphi Process.

**Table 1 ijerph-20-07126-t001:** POCUS clinical domains.

Clinical Domains	Number of Skillsets (*N* = 93)
1. Obstetrics and gynaecology	20
2. Cardiology	10
3. Trauma	4
4. Abdominal	17
5. Vascular	3
6. Musculoskeletal	10
7. Pulmonary	8
8. Ocular	6
9. Procedural	10
10. Ultrasound protocols	5

POCUS = Point-of-care ultrasound.

**Table 2 ijerph-20-07126-t002:** Baseline characteristics of the participants.

Variables	Frequency
**Provincial distribution of panelists**	
Eastern Cape	5
Free State	3
Gauteng	6
KwaZulu-Natal	3
Limpopo	1
Mpumalanga	4
Northern Cape	2
Western Cape	7
North-West	1
No response	4
**Training in ultrasound**	
Postgraduate Diploma in US	1
FPD US courses	7
EMSSA POCUS	6
Other US short courses	9
Informal training in US	2
No response	4
**Experience in patient evaluation with US (years)**	
≤3	8
4–6	8
7–9	6
≥10	10
No response	4

EMSSA = Emergency Medicine Society of South Africa; FDP = Foundation for Professional Development; POCUS = Point-of-care ultrasound; US = Ultrasound.

**Table 3 ijerph-20-07126-t003:** Breakdown of selection per clinical domain.

Clinical Domain	Skillsets (90)	Essential	Optional	Non-Essential
Obstetrics and Gynaecology	20	16	3	1
Cardiology	10	4	6	0
Trauma	4	4	0	0
Abdominal	15	7	8	0
Vascular	3	2	1	0
Musculoskeletal	9	0	4	5
Pulmonary	8	3	3	2
Ocular	6	0	0	6
Procedural	10	6	3	1
Ultrasound protocols	5	4	1	0

**Table 4 ijerph-20-07126-t004:** Consensus on essential POCUS skillsets.

1.1.1	Identification of the presence of an intrauterine pregnancy	Essential	R1
1.1.2	Determination of the viability of an intrauterine pregnancy	Essential	R1
1.1.3	Detection of the fetal heart rate using M-Mode	Essential	R2
1.1.4	First trimester pregnancy gestational age assessment by crown rump length detection	Essential	R1
1.1.5	Recognition of molar pregnancy	Essential	R1
1.2.1	Determination of placental position	Essential	R1
1.2.2	Determination of the fetal presentation	Essential	R1
1.2.3	Performing a gestational age assessment and fetal weight estimation using abdominal circumference (AC), biparietal diameter (BPD), and femoral length (FL).	Essential	R1
1.2.10	Confirmation of fetal death	Essential	R1
1.2.4	Assessment of the amniotic fluid volume using either the four-quadrant calculation or deepest single pocket approach	Essential	R2
1.2.5	Assessment of the placenta for features of placental abruption	Essential	R2
1.2.8	Assessment of fetal well-being using the biophysical profile	Essential	R4
1.2.9	Assessment of fetal well-being during third trimester using umbilical artery doppler	Essential	R5
1.3.1	Confirmation of intrauterine device (IUD) position	Essential	R3
1.3.2	Measurement of endometrial thickness	Essential	R2
1.3.3	Assessment of an adnexal mass: simple, complex, and hemorrhagic cysts	Essential	R5
2.1	Detection of a pericardial effusion	Essential	R1
2.10	Measurement of inferior vena cava (IVC) diameter and collapsibility to approximate volume status	Essential	R5
2.2	Assessment of global left ventricle contractility (hyperdynamic/normal/decreased)	Essential	R4
3.1	Assessment of free fluid in the abdominal cavity	Essential	R1
3.2	Assessment of free fluid around the heart with a sub-xyphoid view	Essential	R1
3.3	Assessment of a pneumothorax	Essential	R2
3.4	Assessment of a hemothorax	Essential	R2
4.1.1	Detection of an abdominal aortic aneurysm	Essential	R3
4.2.1	Assessment of the gallbladder for cholelithiasis	Essential	R2
4.2.2	Assessment for acute cholecystitis	Essential	R2
4.2.3	Assessment for common bile duct (CBD) obstruction (choledocholithiasis)	Essential	R5
4.2.4	Assessment for hepato-splenomegaly	Essential	R5
4.4.1	Assessment of renal size	Essential	R4
4.4.5	Identification of a testicular torsion	Essential	R2
5.1	Identification of lower extremity DVT in low-risk cases with 2-Zone discrimination technique	Essential	R1
5.2	Identification of lower extremity DVT with color Doppler and graded compression of the entire limb	Essential	R3
7.1	Identification of pulmonary oedema	Essential	R5
7.3	Identification of pneumothorax	Essential	R2
7.4	Identification of pleural effusion or hemothorax	Essential	R2
9.1	Thoracentesis	Essential	R2
9.2	Paracentesis	Essential	R2
9.3	Peripheral IV placement	Essential	R2
9.4	Central line placement	Essential	R1
9.7	Foreign body identification and removal	Essential	R2
9.8	Fine-needle aspiration/biopsy	Essential	R2
10.1	FAST/E-FAST: Focused assessment with sonography for trauma	Essential	R1
10.2	RUSH: Rapid ultrasound for shock and hypotension	Essential	R2
10.3	BLUE: Bedside lung ultrasound in emergency	Essential	R2
10.4	CLUE: Cardiac limited ultrasound exam	Essential	R2

## Data Availability

The anonymized data from this study will be available for five years post publication. Access will be granted upon receipt of an accepted proposal and will require a signed data sharing agreement.

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
