# Peer review of "Building Consensus on the Point-of-Care Ultrasound Skills Required for Effective Healthcare Service Delivery at District Hospitals in South Africa: A Delphi Study"

_ijerph, 2023, doi:10.3390/ijerph20237126_

Round 1
Reviewer 1 Report
Comments and Suggestions for Authors
Thank you for your submission on consensus recommendation on skills needed for practitioners of PoCUS in different parts of South Africa. Your work is commendable because it addresses the issue of significance of developing consensus guidelines/recommendations that are specific to the region where clinicians need more relevant guidance based on the local expertise, the disease burden, the kind of patients they deal with and the applicability of those guidelines. Your methodology is appropriate given the circumstances of your geographical peculiarities, although face to face discussions and voting in different rounds of Delphi remain the gold standard.
There are some interesting findings such as lack of importance of MSK and small organ ultrasound perceived by your panelists and rather in depth knowledge requirement of obstetrics and gynecology ultrasound. It must reflect the medical practice most of the panelists are engaged in South Africa.
To improve the manuscript, it would be better to expand the table of skillsets and include optional and non essential skills as well with the respective headings so that the reader will get the full picture in one glance. This will also help in future versions of these consensus recommendations to identify if any of those skills at some stage become important enough to be considered as essential. There would thus be a commentary on how and why that happened.
Reviewer 2 Report
Comments and Suggestions for Authors
I applaud the authors for the thoughtful description of their research. The Delphi process is based upon expert opinion and this study does not utilize "experts" to create the consensus. An expert panel based upon "perceived proficiency" is likely inadequate to formulate a true consensus statement. This issue is discussed in the limitations of the study and the conclusion describes a "list"of POCUS skillsets, not expert consensus. This will likely provide valuable data to inform future research the but conclusion should be more clear and state that though a Delphi process was implemented, no expert opinion could be established due to the lack of using defined experts to complete the process
Reviewer 3 Report
Comments and Suggestions for Authors
The study under review addresses a significant gap in the understanding of the essential Point of Care Ultrasound (POCUS) skillsets required by generalist medical practitioners in South African district hospitals. The authors use a Delphi consensus study to identify these skillsets. Delphi methodology is apt for gathering expert opinions, especially when geographical constraints hinder conventional expert collaboration.
Strengths:
-
Robust Methodology: The use of the Delphi method is appropriate for this study, given the diverse geographical locations of the participants. The iterative rounds allowed for comprehensive exploration of opinions, ensuring a well-rounded consensus.
-
Relevance to Local Context: The study acknowledges the regional variations in healthcare needs, providing a comprehensive understanding of the unique requirements in South African district hospitals. This approach adds depth to the findings.
-
Practical Implications: The study's focus on essential POCUS skillsets has practical implications for medical training, curriculum development, and healthcare policy in South Africa. By tailoring the skillsets to the local context, the study provides actionable insights.
Weaknesses:
-
Limited Participant Training Standardization: The variability in participants' POCUS training levels and experiences might introduce bias into the consensus. The study lacks a standardized measure to assess the participants' expertise, potentially impacting the quality of the consensus.
-
Limited Diversity in Participants: While the study includes experts from diverse geographical locations, it might benefit from a more diverse participant pool, including practitioners from different cultural backgrounds and healthcare systems. This broader representation would enhance the study's applicability to a wider range of contexts.
-
Omitted Skillsets: The study fails to include certain skillsets that are clinically relevant in the local context, such as specific ultrasound protocols for diseases like HIV-associated tuberculosis. This omission raises questions about the comprehensiveness of the skillset list.
-
Lack of Comparative Analysis: The study does not compare the identified skillsets with those recommended in other regions or countries facing similar healthcare challenges. Such a comparison could provide valuable insights into the uniqueness of the South African context.
Conclusion:
The study makes a significant contribution by identifying essential POCUS skillsets for South African district hospitals. The regional focus adds depth to the findings, providing valuable insights for medical training and policy formulation. However, the study could benefit from a more standardized approach to participant selection and a comparative analysis with global practices. Addressing these limitations would enhance the study's impact and applicability in the broader context of global healthcare.
Please improve:
-
-
Introduction:
- Provide a more detailed background on point of care ultrasound (POCUS) and its importance in healthcare. Explain briefly why it is regarded as the 'stethoscope of the future'.
- Clearly state the research question or objective of the study in the introduction section.
-
Methods:
- Provide a concise summary of the methodology used for the Delphi process. Explain why the Delphi method was chosen and its advantages for this study.
- Describe the criteria for expert selection in more detail. Explain why these experts were considered suitable for the study.
- Clarify the criteria used to categorize skillsets as essential, optional, and non-essential.
-
-
Discussion:
- Compare the results of this study with existing literature on POCUS skillsets, both in South Africa and globally. Discuss any similarities or differences.
- Address any limitations of the study, such as the attrition rate of participants, and explain how these limitations might have influenced the results.
- Discuss the implications of the findings for medical training in South Africa and potentially other regions with similar healthcare contexts.
- Consider discussing the potential challenges and solutions for implementing the identified skillsets in district hospitals.
-
Conclusion:
- Provide a concise summary of the key findings without introducing new information.
- Highlight the significance of the study's findings and their potential impact on healthcare practice and training.
-
-
Additional Considerations:
- Consider adding a brief section on future research directions, suggesting areas where further studies could be conducted based on the current findings.
The quality of the English language in the provided text is generally good. The sentences are well-structured, and the vocabulary used is appropriate for an academic context. However, there are a few areas where improvements can be made to enhance clarity and readability:
-
Complex Sentence Structure: Some sentences are quite complex and might be challenging for readers to follow. Simplifying these sentences can enhance comprehension.
-
Word Choice: In a few instances, there are words or phrases that could be replaced with more precise or commonly used terms to improve clarity. For example, in the sentence "The implications of these findings for the healthcare sector are paramount," the word "paramount" could be replaced with a simpler term like "crucial" for clearer understanding.
-
Redundancy: There are a few instances of redundant phrases, such as "in order to" which can be simplified to "to." For example, "in order to enhance the overall readability" can be written as "to enhance the overall readability."
-
Conciseness: Some sentences could be made more concise without losing their meaning. For example, the sentence "Consider adding a brief section on future research directions, suggesting areas where further studies could be conducted based on the current findings" could be simplified to "Consider adding a section on future research directions, suggesting areas for further studies based on the current findings."
